# Herpes Simplex Virus Type 1 Induces AD-like Neurodegeneration Markers in Human Progenitor and Differentiated ReNcell VM Cells

**DOI:** 10.3390/microorganisms11051205

**Published:** 2023-05-04

**Authors:** Blanca Salgado, Isabel Sastre, Maria J. Bullido, Jesus Aldudo

**Affiliations:** 1Centro de Biologia Molecular “Severo Ochoa” (C.S.I.C.-U.A.M.), Universidad Autonoma de Madrid, 28049 Madrid, Spain; bsalgado@cbm.csic.es (B.S.); isastre@cbm.csic.es (I.S.); 2Centro de Investigacion Biomedica en Red de Enfermedades Neurodegenerativas (CIBERNED), Instituto de Salud Carlos III, 28031 Madrid, Spain; 3Instituto de Investigación Sanitaria del Hospital Universitario La Paz—IdiPAZ, 28046 Madrid, Spain

**Keywords:** Alzheimer’s disease, HSV-1, ReNcell VM, differentiation, infection, neurodegeneration, lysosome alterations, neurons, glial cells

## Abstract

An increasing body of evidence strongly suggests that infections or reactivations of herpes simplex virus type 1 (HSV-1) may be closely linked to Alzheimer’s disease (AD). Promising results have been obtained using cell and animal models of HSV-1 infection, contributing to the understanding of the molecular mechanisms linking HSV-1 infection and AD neurodegeneration. ReNcell VM is a human neural stem cell line that has been used as a model system to study the impact of various infectious agents on the central nervous system. In this study, we demonstrate the suitability of the ReNcell VM cell line for developing a new in vitro model of HSV-1 infection. By following standard differentiation protocols, we were able to derive various nervous cell types, including neurons, astrocytes, and oligodendrocytes, from neural precursors. Additionally, we demonstrated the susceptibility of ReNcell VM cells, including precursor and differentiated cells, to HSV-1 infection and subsequent viral-induced AD-like neurodegeneration. Our findings support the use of this cell line to generate a new research platform for investigating AD neuropathology and its most significant risk factors, which may lead to important discoveries in the context of this highly impactful disease.

## 1. Introduction

Alzheimer’s disease (AD) is the most common form of dementia worldwide. Given the ageing global population, the incidence of AD and the economic burden on health systems are expected to increase in the following decades, reaching 152 million patients by 2050 [1]. Despite the genetic nature of a minority of cases, the most prevalent form of AD (sporadic AD (sAD)) is thought to have a multifactorial aetiology that remains elusive [2]. In addition to ageing, in recent times other modifiable or targetable risk factors for sAD have gained interest [3], including infectious agents, such as herpes simplex virus-1 (HSV-1). HSV-1 is a ubiquitous and prevalent neurotropic DNA virus that causes mucosal lesions and, in more severe cases, encephalitis. HSV-1 belongs to the Alphaherpesvirinae subfamily, which can establish latent infections in sensory ganglia. After primary replication in the mucosal epithelium, the virus becomes latent in neurons of the peripheral nervous system and can periodically be reactivated by diverse stimuli. Moreover, HSV-1 can reach the central nervous system, and its repeated reactivation has been postulated to be involved in the pathogenesis of AD [4,5].

Over the years, researchers have identified multiple connections between AD and HSV-1 infection. Numerous studies have demonstrated that HSV-1 induces AD-related pathological changes contributing to neuronal loss and brain atrophy, such as neuroinflammation and oxidative stress, autophagy and lysosomal impairment, tau hyperphosphorylation, accumulation of intracellular β-amyloid (Aβ), blood–brain barrier disruption, and DNA damage, as reviewed in [6]. Other pieces of evidence supporting this relationship are the presence of HSV-1 in AD brains, links between ε4 APOE allele and susceptibility to infection [7,8], and the positive effects of certain antivirals on AD development [9]. Finally, recent studies by our research group have revealed that cholesterol homeostasis and lysosome pathway, both processes altered in AD [10,11], could play a role in the neurodegeneration induced by HSV-1. Specifically, we observed a reduction in the lysosomal activity in HSV-1-finfected cells and a strong impact of several lysosomal-related genes on HSV-1-induced neurodegeneration [12,13,14]. Overall, these data evidence the relevance of models based on HSV-1 infection as a powerful tool to explore mechanisms potentially involved in AD pathogenesis. 

Considering the limitations and ethical concerns arising from the use of animals for experimental purposes, there is an imperative need to develop new models for sAD research. Neural stem cells (NSCs) have been proposed as potential tools for research and replacement therapy in neurodegenerative diseases such as AD [15]. Moreover, the ability to integrate the advantages of stem-cell-based cellular models with genome editing techniques and engineered three-dimensional (3D) microenvironments holds tremendous potential for developing in vitro systems to explore cellular responses to infections that are both relevant and predictive [15,16].

ReNcell VM is a cell line of human NSCs derived from the ventral mesencephalic region of the developing human brain and immortalised by retroviral transduction with the *v-myc* oncogene [17]. These cells exhibit the remarkable ability to self-renew and differentiate into neurons, astrocytes, and oligodendrocytes, which are the three main cell lineages in the central nervous system [18], allowing for the recapitulation of essential intercellular interactions in vitro. One of the key advantages of ReNcell VM cells is their ability to maintain their pluripotency, even after long-term culture in vitro, allowing for the development of stable cell lines with consistent properties [17]. This makes them useful for large-scale screening assays and for the study of genetic and environmental factors that affect neural development. Moreover, ReNcell VM has been demonstrated to be a suitable tool for modelling AD and other neurological disorders in both two-dimensional (2D) and 3D culture systems. This allows for the development of 3D models that more accurately replicate the spatial complexity of the brain environment during the disease [19,20,21]. Furthermore, these cells have been used to investigate the effects of various infectious agents on the nervous system [22,23]. Overall, the ReNcell VM is a valuable tool for studying the impact of infectious agents on the development of nervous system pathologies and for understanding the mechanisms underlying these interactions.

Here, we aimed to develop a new model of HSV-1 infection using ReNcell VM cells. To achieve this goal, we characterised the differentiation of ReNcell VM cells and analysed the effects of HSV-1 infection at distinct stages of differentiation. Our findings indicate that ReNcell VM cells are permissive to HSV-1 infection and suggest that the virus is capable of inducing an AD-like phenotype in these cells. Thus, the establishment of this novel experimental system will enable us to study the pathophysiological mechanisms linking AD with HSV-1 infection.

## 2. Materials and Methods

### 2.1. Cell Culture and Differentiation

The human neural stem cell line ReNcell VM was kindly provided by Dr. A. Cuadrado [24]. ReNcell VM cells were cultured on Corning Matrigel hESC-Qualified Matrix (Corning, Corning, NY, USA)-coated plates with proliferation medium: neurobasal medium (Gibco, Waltham, MA, USA) supplemented with 2% (*v/v*) B27 supplement (Gibco), 2 mM glutamax (Gibco), 50 μg/mL gentamicin, 20 ng/mL b-FGF (Basic Fibroblast Growth Factor; Peprotech, Waltham, MA, USA), and 20 ng/mL EGF (Epidermal Growth Factor; Peprotech), as described in [24]. Cells were cultured at 37 °C in a 5% CO_2_ atmosphere, and cell passages were performed every 3–4 days. 

At 80–90% confluence, differentiation was induced by growth factor withdrawal from the proliferation medium and monitored using phase-contrast microscopy and expression analysis of neuronal precursor (Nestin, SOX-2, and Ki-67 (MKI67)), neuronal (β-tubulin III, synapsin I (SYN1), and synaptophysin (SYP)), dopaminergic (tyrosine hydroxylase (TH) and dopamine decarboxylase (AADC)), and glial markers (glial fibrillary acidic protein (GFAP) and oligodendrocyte transcription factor (OLIG2)). Meanwhile, the differentiation medium was changed every 2–3 days. 

### 2.2. HSV-1 Infection

At 70–75% confluence, cell cultures were infected at different multiplicities of infection (moi) with wild-type HSV-1 strain KOS 1.1 (kindly provided by Dr. L. Carrasco). This strain was obtained, propagated, and purified from Vero cells, as described in [25]. Cells were incubated in a viral solution for 1 h at 37 °C. Then, the unbound virus was removed, and the cells were incubated in culture medium at 37 °C until their collection. Control samples (*mock*) were incubated in virus-free suspensions. Viral titres in cell culture supernatants were determined by plaque assays [26].

### 2.3. Viral DNA Quantification

DNA was purified using the QIAamp^®^ DNA Blood Kit (QIAGEN, Hilden, Germany). The amount of HSV-1 DNA was quantified using real-time quantitative PCR with an CFX-384 Real-Time PCR System (BioRad, Hercules, CA, USA) with a custom designed TaqMan assay specific for the *US12* viral gene (forward primer: 5′-CGTACGCGATGAGATCAATAAAAGG-3′; reverse primer: 5′-GCTCCGGGTCGTGCA-3′; TaqMan probe: 5′′-AGGCGGCCAGAACC-3′). Viral DNA content was normalised in terms of human genomic DNA and quantified with a predesigned TaqMan assay specific for the *18S* (Hs9999991_s1; Applied Bio systems, Waltham, MA, USA). The quantification data are expressed as the viral DNA copy number per ng of genomic DNA. 

### 2.4. Immunofluorescence Analysis

Cells grown on coverslips were fixed in 4% paraformaldehyde (PFA) and permeabilised with blocking solution (2% horse or foetal calf serum, 0.2% Triton X-100 in phosphate buffer saline (PBS) pH 7.4). Then, coverslips were incubated with primary antibodies and with Alexa Fluor-coupled secondary antibodies diluted in blocking solution (Table 1). Finally, cells were counterstained with 4,6-diamino-2-phenylindole (DAPI) (Merck, Rahway, NJ, USA) in PBS and mounted on microscope slides using Mowiol medium (Sigma-Aldrich, St. Louis, MI, USA). The overall procedure was performed at room temperature (RT), and the samples were protected from light. Sample visualisation was performed in the FRET inverted microscope Axiovert200 (Zeiss, Jena, Germany) coupled to a monochrome CCD camera and in an LSM 900 laser scanning confocal microscope (Zeiss) coupled to a vertical Axio Imager 2 vertical microscope (Zeiss). Immunofluorescence images were obtained using Metamorph or ZEN Blue 3.4 imaging software and processed with Adobe Photoshop software (San Jose, CA, USA).

### 2.5. Cell Lysates and Western Blot Analysis

Lysates were obtained by incubating the cell samples in the radioimmunoprecipitation assay (RIPA) buffer (10 mM Tris-HCl pH 7.5, 50 mM NaCl, 1% Nonidet P-40, 0.2% sodium deoxycholate, 0.1% sodium dodecyl sulphate (SDS), and 1 mM EDTA) containing protease (Complete^TM^, Mini, EDTA-free Protease Inhibitor Cocktail, Roche, Basel, Switzerland) and phosphatase (PhosSTOP^TM^, Roche) inhibitors. Before the Western blot analysis, the protein concentrations in the cell lysates were determined by Bicinchoninic acid assay (BCA, Pierce, Waltham, MA, USA) following the manufacturer’s instructions. Protein separation was performed using Laemmli discontinuous SDS-polyacrylamide gel electrophoresis and Laemmli loading buffer (25 mM Tris-HCl pH 6.3, 10% glycerol, 2% SDS, 5% beta mercaptoethanol, and 0.01% bromophenol blue). After electrophoresis and transference to a nitrocellulose membrane, the membranes were blocked with PBS—3%, BSA—0.2%, Tween 20, or PBS—5% milk—0.2% Tween 20 solution. Incubations with primary and peroxidase-coupled secondary antibodies (Table 1) diluted in dilution buffer (PBS—1%, BSA—0.05%, Tween 20, or PBS—1% milk—0.05% Tween 20) were performed at RT. Finally, detection by enhanced chemiluminescence was carried out using ECL Western blotting detection reagents (Amersham Biosciences, Amersham, UK) according to the manufacturer’s instructions. To quantify the intensity of the protein bands, densitometric analysis was performed using Image Lab software (Bio Rad, Hercules, CA, USA).

**Table 1 microorganisms-11-01205-t001:** List of antibodies used in Western blot (WB) and immunofluorescence (IF) analysis.

Antibodies		Dilutions		Reference
		WB	IF	
Progenitors	Nestin		1/200	Biolegend: 656801
	SOX2		1/100	Abcam: ab79351
Neurons	Beta III tubulin		1/100	Abcam: ab18207
Astrocytes	GFAP		1/100	Biolegend: 644701
Oligodendrocytes	OLIG2		1/200	Milipore: AB9610
Viral markers	ICP4	1/1000	1/100	Abcam: ab6514
	UL42	1/1000		Santa Cruz (13C9) sc-53331
	gC	1/3000	1/300	Abcam: ab6509
	gB/gD		1/300	Provided by Dr. Enrique Tabares (UAM)
Neurodegeneration markers	Aβ40		1/100	Invitrogen 44348A
	Aβ42		1/100	Invitrogen 44-344
	p-Tau Thr205	1/250	1/50	Invitrogen 44-738G
	p-Tau Ser422	1/250	1/50	Invitrogen 44-764G
	AT8		1/50	Thermo Fisher MN1020
	LC3B	1/500	1/100	Sigma L7543
Housekeeping proteins	GAPDH	1/1000		Santa Cruz sc-51906
	α-Tubulin	1/10,000		Sigma T5168
Secondary antibodies	Anti-mouse-POD	1/25,000		Vector PI-2000
	Anti-rabbit-POD	1/25,000		Nordic GAR/IgG (H+L)/PO
	Alexa-555 anti-mouse		1/1000	Thermo Fisher A-21137
	Alexa-488 anti-rabbit		1/1000	Thermo Fisher A-21206

### 2.6. Quantitative RT-PCR

The mRNA transcribed from each gene was quantified by reverse transcription followed by real-time PCR. Briefly, total RNA was isolated with a QIAamp^®^ RNA Blood Mini Kit (QIAGEN) and subjected to reverse transcription using the High Capacity RNA-to-cDNA Kit (Applied Bio systems, Waltham, MA, USA). cDNAs were amplified using PCR with primers specific for several progenitor, neuronal, dopaminergic, and glial genes, as previously described [27,28] (Table 2). The data were normalised with respect to the value for the *β-actin* gene due to the fact of its constant expression. Real-time PCR assays were performed in a CFX-384 Real-Time PCR System (*BioRad*). The quantities of mRNAs were determined using Bio Rad CFX maestro 2.2. software.

### 2.7. Secreted Aβ Measurements

Conditioned media from mock-infected and infected samples were assayed for human Aβ40 and Aβ42 using commercial sandwich enzyme-linked immunosorbent assay (ELISA) kits (Wako, Tokyo, Japan) according to the manufacturer’s instructions. First, media were collected and inactivated with UV exposure. After centrifugation, the samples were kept at −70·°C. Once frozen, they were concentrated 10-fold by lyophilisation and suspension in PBS with a protease inhibitor cocktail (Roche). The bound detection anti-Aβ antibody produced a colorimetric signal that was read at 450 nm. The absolute values for Aβ40 and Aβ42 are expressed as picomoles per litre of incubation medium (pM).

### 2.8. Quantification of Lysosome Load

The lysosome load was determined using the acidotropic probe LysoTracker Red DND-99 (LTR, Thermo Fisher, Waltham, MA, USA), which freely passes through cell membranes and typically concentrates in acidic organelles. One hour before the end of the treatments, cells were exposed to 0.5 μM LTR for one hour at 37 °C in culture medium and then washed with PBS. Then, cells were lysed with RIPA buffer and centrifuged at 13,000 g for 10 min. The protein concentration of the lysates was quantified using the BCA method, and the LTR fluorescence of the cell lysates was recorded using a FLUOstar OPTIMA microplate reader (BMG LABTECH, Saitama, Japan) (excitation wavelength: 560 nm; emission wavelength: 590 nm).

### 2.9. Cathepsin Activity Assays

The enzymatic activity of different cathepsins was determined as previously described with minor modifications [29]. Briefly, ReNcell VM cells were lysed with shaking in 50 mM sodium acetate (pH 5.5), 0.1 M NaCl, 1 mM EDTA, and 0.2% Triton X-100. Lysates were clarified by centrifugation and immediately used for the determination of proteolytic activity. A total of 50–100 μg of protein from the cell lysates were incubated for 30 min in the presence of the following fluorogenic substrates (all from Enzo Life Sciences, Farmingdale, NY, USA): Z-VVR-7-amino-4-methyl-coumarin (AMC) (P-199; most sensitive substrate for cathepsin S; 20 mM), and the cathepsin D/E fluorogenic substrate Mca-GKPILFFRLK (Dnp)-D-Arg-NH2 (P-145; 10 mM). The fluorescence released was quantified with a microtiter plate reader (*Tecan Trading AG*) with excitation at 360 nm and emission at 430 nm (Z-VVR-AMC) or 320 nm and 400 nm (cathepsin D and E fluorogenic substrate).

### 2.10. Statistical Analysis

Differences between groups were analysed pairwise using the two-tailed Student’s t-test, or the one-sample t-test in the case of data expressed as relative values. The significance was recorded at *p* < 0.05 (*), *p* < 0.01 (**), and *p* < 0.001 (***). The statistical analyses were performed using Microsoft Excel and GraphPad Software (Redmond, DC, USA).

## 3. Results

### 3.1. ReNcell VM Cells Differentiate into Neuronal and Glial Cells

ReNcell VM is a human neural stem cell line derived from the ventral mesencephalic region of the developing human brain. This cell line displays a polygonal morphology and a cobblestone-like growth in the presence of growth factors. According to previous studies, the withdrawal of growth factors from the proliferation medium leads to the differentiation of ReNcell VM cells into neurons and glial cells [17]. We confirmed this capacity using phase-contrast microscopy, immunofluorescence assays, and gene expression analysis (as shown in Figure 1). Following the induction of differentiation, proliferation halted, cells began to elongate, and axon-like structures started to grow from cell bodies. Morphological changes become evident on the second day of differentiation and were most pronounced by the fourth day, where the neuronal and glial cell morphology was clearly observed by phase-contrast microscopy. Our longest differentiation experiment lasted 21 days, which provides evidence of the high stability and extended survival of ReNcell VM neuronal cultures (Figure 1A).

Immunofluorescence assays revealed that undifferentiated cells showed a positive signal for the neural stem cell markers Nestin and SOX2 and confirmed that ReNcell VM cells differentiate into all three neural lineages: neurons, astrocytes, and oligodendrocytes (Figure 1B). The most abundant cell types were neurons and astrocytes, with similar percentages, whereas oligodendrocytes and catecholaminergic (TH+) neurons were scarce in these cultures (<10%). These percentages are consistent with previously published data, which show that differentiation resulted in approximately 50% glial (primarily astrocytic) cells and 50% neurons of which 10% had a dopaminergic phenotype [30]. The expression patterns of the astrocytic marker GFAP and the neuronal marker βIII-tubulin did not show co-staining of both markers within the same cell, indicating the proper functioning of the differentiation process (Figure 1C). Finally, the gene expression analysis of several markers for progenitor, neurons, and glial cells was performed using RT-qPCR. While the expression of Ki-67, a cell proliferation marker highly expressed in NSCs, strongly decreased during differentiation and the expression of markers for neurons, dopaminergic cells, and glial cells increased (Figure 1D). In summary, our results demonstrate an irreversible conversion of ReNcell VM neural stem cells into a post-mitotic neuronal population.

### 3.2. ReNcell VM Cells Are Permissive to HSV-1 Infection

Our first aim was to determine whether ReNcell VM cells are permissive to HSV-1 infection and to establish the infection conditions for subsequent experiments. In this cell model, HSV-1 establishes a lytic replication cycle resulting in cell detachment and lysis ~20–22 h after infection. The analysis of the expression of several viral proteins in the ReNcell VM cells exposed to different viral doses was performed with Western blot. We found that the HSV-1 immediate early protein ICP4, the early protein UL42, and the true-late glycoprotein gC were robustly expressed at 18 h post-infection (hpi) and remained constant at the highest viral doses tested (Figure 2A). Next, immunofluorescence assays using an anti-gC antibody were performed to determine the infection rate. After exposing cells to HSV-1 at an moi of 3 for 8 h, we estimated that approximately 70% of the cells were positive for gC staining. This percentage increased to almost 100% at 18 hpi (Figure 2B). Finally, we investigated whether ReNcell VM cells can support productive HSV-1 infection. The analysis of HSV-1 genome replication using quantitative PCR showed a dose-dependent increase in the copies of viral DNA, confirming the ability of HSV-1 to replicate in ReNcell VM cells (Figure 2C). Viral titre assays also showed a dose-dependent increase in infectious viral particles in the culture supernatants of infected ReNcell VM cells, indicating the ability of HSV-1 to complete the lytic cycle in these cells (Figure 2D).

Taking these data into account, all subsequent infections were performed at an moi of 1 and 3 pfu/cell for 18 h, as these infection conditions ensure both the accumulation of viral proteins and the infection of almost all cells.

### 3.3. HSV-1 Induces AD-like Neurodegeneration Markers in ReNcell VM Cells

One of the primary features of AD is the build-up of Aβ extracellular aggregates, which form senile plaques. Previous studies have shown that HSV-1 infection can increase the accumulation of intracellular Aβ and hinder the non-amyloidogenic pathway, leading to a decrease in Aβ secretion [31]. Immunofluorescence was used to determine whether HSV-1 infection modifies Aβ levels in ReNcell VM cells. In the mock-infected cells, none of the anti-Aβ antibodies tested were able to detect intracellular Aβ. When the cells were exposed to HSV-1, a strong accumulation of Aβ took place, as revealed by immunofluorescence using antibodies specific to Aβ40 and Aβ42. In fact, the accumulation of Aβ was detected in the early stages of infection and was evident at 3 hpi for Aβ40 and 5 hpi for Aβ42 (Figure 3A). Finally, the Aβ content of extracellular media from ReNcell VM cells was determined by ELISA. When these cells were infected with HSV-1, the Aβ40 and Aβ42 levels fell drastically (Figure 3B). Our data suggest that the inhibition of Aβ secretion could be a mechanism causing intracellular Aβ accumulation in HSV-1-infected ReNcell VM cells.

Tau is a microtubule-associated protein that is hyperphosphorylated in AD brains, leading to the formation of neurofibrillary tangles (NFTs), another important neuropathological feature of the disease. We performed immunofluorescence analysis to determine whether HSV-1 infection modifies the phosphorylation state of tau in ReNcell VM cells using antibodies that recognise different phosphorylated epitopes of tau characteristic of NFTs: Ser422 and Thr205. In noninfected cells, phosphorylation-sensitive antibodies weakly stained the cytoplasm. Notably, stronger immunoreactivity was observed in mitotic cells, which is consistent with the results of other authors, who reported that the abnormal tau-phosphorylation characteristic of AD also occurs during mitosis [32]. Following exposure to HSV-1, a dot-pattern staining of phosphorylated tau became visible at 3 hpi. Furthermore, a strong accumulation of phosphorylated tau was observed in the infected cells at 18 hpi (Figure 3C). Subsequently, we employed Western blot analysis to quantify the effects of HSV-1 infection on tau phosphorylation. Consistent with the results of the immunofluorescence experiments, we observed a marked increase in phosphorylated tau levels at both of the examined phosphorylation sites in a viral dose-dependent manner (Figure 3D).

Taken together, these findings are consistent with those obtained in previous reports from our lab in human neuroblastoma cells, confirming that HSV-1 infection strongly modifies the APP processing and tau phosphorylation state in neuronal cells [26,33].

### 3.4. Lysosomal Pathway Is Impaired by HSV-1 in ReNcell VM Cells

Another neuropathological feature of AD induced by HSV-1 is the dysfunction of the autophagy–lysosome pathway. Among those alterations, increases in LC3 lipidation have been previously reported. Upon the activation of autophagy, LC3 is converted from its cytosolic form, LC3-I, to the autophagic membrane-bound form LC3-II. LC3-II binds specifically to autophagic membranes and remains membrane bound throughout the pathway [34]. Immunofluorescence studies showed that LC3 accumulated in HSV-1-infected ReNcell VM cells, whereas it was almost undetectable in noninfected cells. The accumulation of LC3 was detectable as early as 3 hpi, while 18 h infections led to the accumulation of LC3 in almost all infected cells (Figure 4A). The punctate and concentrated staining of LC3 was consistent with its localisation on autophagic compartments. These results were confirmed by Western blot analysis of LC3 levels, which showed a viral dose-dependent increase in LC3-II, accompanied by a reduction in LC3-I levels (Figure 4B). Taken together, these findings suggest that the autophagic response is altered during HSV-1 infection, leading to the accumulation of autophagosomes in ReNcell VM cells.

The effect of HSV-1 on the quantity of lysosomes was also explored using the lysosomotropic probe LysoTracker Red (LTR). As this probe concentrates in acidic organelles, measuring the fluorescence of LTR is a widely used technique for quantifying the cellular load of lysosomes. HSV-1 infection induced a significant decrease in the LTR fluorescence levels of ReNcell VM cells (Figure 4C), indicating potential alterations in lysosomal function. To investigate this further, the activity of lysosomal cathepsins was assessed using fluorogenic substrates specific for cathepsins S, D, and E (Figure 4D). The results show that the cathepsin activities in ReNcell VM cells exposed to HSV-1 were significantly lower than in noninfected cultures, indicating that the infection induces a defect in the proteolytic activity of lysosomes.

### 3.5. Differentiated ReNcell VM Cells Are Permissive to HSV-1 Infection

To characterise the infection in neuronal cultures, ReNcell VM cells were differentiated for 8 days and then infected with HSV-1. Phase-contrast microscopic images revealed that neuronal integrity was maintained but degeneration of neurites began to be observed in cells infected for 24 h (Figure 5A). Therefore, differentiated ReNcell VM cells seem less susceptible to the virus than progenitor cells. Immunofluorescence study using an antibody specific to the viral protein ICP4 demonstrated the expression of ICP4 in the nuclei of infected cells. ICP4 is an essential protein of the viral replication compartments (VRCs) that is involved in viral gene expression and DNA replication. Additionally, the viral glycoproteins gB, gD, and gC were observed to accumulate in the infected cells. (Figure 5B). Differentiated ReNcell VM cell cultures showed over 90% of cells infected at an moi of 3 for 18 h, making them ideal conditions for subsequent studies on virus-induced neurodegeneration. These results agreed with those obtained by Western blot analysis, which show the accumulation of immediate early (ICP4), early (UL42), and true late (gC) viral proteins in infected cells (Figure 5C). Moreover, the analysis of HSV-1 genome replication using quantitative PCR confirmed the ability of HSV-1 to replicate in differentiated ReNcell VM cells (Figure 5D). Finally, viral titre assays revealed the presence of infectious viral particles in the culture supernatants of HSV-1-infected differentiated ReNcell VM cells. The number of viral particles in the differentiated cells was lower than that observed in the progenitor cells. In addition, the viral particles were detectable at longer infection times. These findings suggest that HSV-1 can complete the lytic cycle in differentiated neuronal cultures and also confirm that differentiated cells are less susceptible to the virus than progenitor cells (Figure 5E).

### 3.6. HSV-1 Induces an AD-like Phenotype in Differentiated ReNcell VM Cells

Since HSV-1 infection induced the appearance of AD-like neurodegeneration markers in progenitor ReNcell VM cells, we wanted to verify if this phenotype could also be replicated in differentiated neuronal cultures. After 8 days of differentiation, ReNcell VM cells that were infected with HSV-1 displayed intracellular accumulation of Aβ40 and Aβ42 peptides (Figure 6A). Furthermore, an ELISA analysis of secreted Aβ40 and Aβ42 demonstrated a significant decrease in the levels of both peptides following infection (Figure 6B). Next, we assessed whether tau phosphorylation was affected in differentiated ReNcell VM cells. Our immunofluorescence experiments revealed that infected cells accumulated hyperphosphorylated tau, as detected with antibodies specific for the phosphorylated epitopes of tau AT8 and ser422 (Figure 6C). This finding was subsequently confirmed by Western blot analysis using an antibody that recognises the phosphorylated epitope thr205 (Figure 6D). Taken together, our findings indicate that the changes in the Aβ levels and tau phosphorylation state that were found in infected ReNcell VM cells were also observed in differentiated neuronal cultures.

Finally, we investigated the impact of HSV-1 infection on the autophagy–lysosome pathway in differentiated ReNcell VM cells. Immunofluorescence images revealed the accumulation of LC3 dots in infected cells (Figure 7A), indicating an accumulation of autophagosomes induced by the virus, as also observed in progenitor ReNcell VM cells. In contrast to the effects found in progenitor cells, the fluorescence levels of LTR were not significantly altered in differentiated cells, suggesting that the infection did not affect lysosomal burden (Figure 7B). However, when monitoring the activity of cathepsins S, D, and E, we observed a reduction in the enzymatic activity of all tested cathepsins (Figure 7C), suggesting that HSV-1 induces a defect in the proteolytic activity of lysosomes in differentiated ReNcell VM cells.

In summary, our findings suggest that HSV-1 infection is capable of inducing an AD-like phenotype in ReNcell VM neuronal cultures, reproducing the behaviour observed in progenitor ReNcell VM cells.

## 4. Discussion

The increasing prevalence of AD underscores the need for new experimental models that accurately recapitulate the complexity of this human disorder and overcome the limitations of current study platforms. Here, we propose the development of a new model of HSV-1 infection and neurodegeneration using ReNcell VM cells. This cell line has several advantages as a model system for studying infections of the nervous system. First, ReNcell VM cells are easily cultured in vitro, allowing for high-throughput screening and large-scale experiments [17]. Second, ReNcell VM is a human neural stem cell line derived from the ventral mesencephalon and can be differentiated into distinct cellular types of the human nervous system [17,18], making them a relevant model for human diseases affecting the nervous system. Finally, ReNcell VM cells can be genetically manipulated, allowing for the study of specific aspects of the host response to infection [35,36]. In this work, we have shown that ReNcell VM cells can be differentiated into neurons, astrocytes, and oligodendrocytes and that this differentiation process is irreversible. These findings are significant, as they provide us with a reliable and stable source of post-mitotic neuronal populations for future research.

The ReNcell VM cell line has been used to study the interactions between the immune system and the nervous system during infection. For instance, Zika virus (ZIKV) infection, associated with severe neonatal microcephaly, has been described to induce pyroptosis, a form of cell death associated with inflammatory responses, in ReNcell VM cells. In addition, ZIKV is able to cause neurosphere pyroptosis and impair the growth and morphogenesis of healthy neurospheres derived from ReNcell VM cells [37]. Apart from their use in basic research, neural stem cells have shown promise in preclinical studies as a source of cells for cell-based therapies for neurodegenerative diseases. For example, implantation of two c-myc immortalised human mesencephalic-derived clonal cell lines, which are very similar to the ReNcell VM cell line, ameliorated behavioural dysfunction in a rat model of Parkinson’s disease [38]. Thus, the ReNcell VM human neural progenitor cell line is a valuable tool for advancing our understanding of the biology of neurological diseases and the impact of infectious agents on the nervous system, as well as for understanding the mechanisms underlying these interactions.

The experiments performed in the present work confirmed the ability of ReNcell VM cells to differentiate into distinct neural cell types and their susceptibility to HSV-1, encouraging their use for the establishment of a new in vitro model of HSV-1 infection and neurodegeneration. Despite the fact that no model of HSV-1 infection has been developed in this cell line yet, previous studies on the effects of HSV-1 on neurogenesis and neurodevelopmental disorders have reported the higher susceptibility to the virus of NSCs compared to their differentiated counterparts, supporting our own conclusions [39,40]. Other authors have proven the suitability of ReNcell VM cells in the study of the mechanisms involved in AD and have reported the promotion of different neuropathological alterations by several inductors, such as DNA damage or familiar AD (fAD) mutations [19,41]. In this line, we confirmed that HSV-1 infection is able to induce certain features of AD-like neurodegeneration in both progenitors and differentiated ReNcell VM cells, such as the inhibition of Aβ secretion, accumulation of intracellular Aβ and hyperphosphorylated tau, and lysosomal alterations. Our data suggest that the strong reduction in extracellular Aβ levels and the concomitant increase in intracellular Aβ in HSV-1-infected ReNcell VM cells could be a consequence of the inhibition of Aβ secretion, resulting in the increased accumulation of Aβ inside the cell. Numerous reports indicate that the lysosome is responsible for degrading Aβ. Therefore, the alteration of lysosomal function induced by HSV-1 could also contribute to the intracellular accumulation of the peptide. In this context, the importance of intracellular Aβ as a key player in Alzheimer’s disease has been highlighted by an increasing number of studies in recent years. It has been reported that intracellular Aβ can drive neuroinflammation and affect pathways involved in cellular stress, synaptic plasticity, axonal transport, and receptor function [42]. These findings reinforce the involvement of HSV-1 in AD pathogenesis. 

The AD-like phenotype appears early after infection in ReNcell VM cells, which reproduces the observations performed in other cell lines [26,33] and suggests that these alterations are independent of viral replication. Moreover, we observed that HSV-1 promotes neurodegeneration not only in neurons but also in GFAP-positive cells. Together with other glial cells, astrocytes are also susceptible to HSV-1 infection, undergoing changes that contribute to neuronal infection and degeneration (reviewed in [4]). Along this line, a model combining different cell types may allow for a major resemblance to the neuropathology induced by HSV-1 via integrating intercellular interactions during infection.

Regarding alterations in the autophagy–lysosome pathway induced by HSV-1, previous studies have reported the accumulation of LC3-II, which is known to be associated with impairments in the autophagosome–lysosome fusion process contributing to loss of proteostasis and neurodegeneration [43], as well as inhibition of cathepsin activity in human neuroblastoma cells infected with HSV-1 [14]. However, in contrast to other cell lines where an accumulation of lysosomes was observed [14], viral infection in ReNcell VM seemed to reduce the lysosomal load. Our observations strongly support that HSV-1 disrupts the autophagy–lysosome pathway. As reviewed in [44], the interplay between autophagy and HSV-1 is a highly complex and poorly understood phenomenon, which may show different outcomes depending on the cell type. There are several potential mechanisms that may account for the observed decrease in lysosomal load. One possible mechanism is the lysosomal deacidification induced by HSV-1, which may inhibit lysosomal hydrolase activity, as seen in our cellular model. This phenomenon has been observed in lysosomal alterations caused by various coronaviruses [45]. Another mechanism by which HSV-1 may modulate lysosomal biogenesis involves altering the levels or activity of the transcription factor TFEB, a master activator of lysosomal biogenesis. This alteration could lead to a decrease in lysosomal levels. Coxsackievirus B3, a virus associated with myocarditis and meningoencephalitis In children, targets TFEB for proteolytic processing to disrupt host lysosomal function by affecting lysosomal biogenesis signalling [46]. Lastly, HSV-1 may impact lysosomal integrity by interacting with lysosomal membrane proteins through its viral proteins. For instance, the influenza A virus reduces the number of lysosomes by interacting with the lysosomal membrane protein LAMP2 through its neuraminidase protein, ultimately causing lysosomal membrane rupture [47]. This is noteworthy because we previously reported that LAMP2 is functionally involved in HSV-1-induced neurodegeneration [12]. In this context, we are currently exploring the role of LAMP2 in the lysosomal alterations induced by HSV-1.

In conclusion, these results are promising and support the feasibility of using ReNcell VM cells as a model for the study of connections between HSV-1 and AD. First, the ability of these cells to differentiate into neuronal and glial cells is of special relevance, considering the generation of 3D models as a future goal. Second, their susceptibility to HSV-1 infection and the neurodegeneration induced by the virus support its use as a platform for the study of the interplay between HSV-1 and AD pathogenesis, as well as other features of neurodegeneration. The previous establishment of 3D models of ReNcell VM cells capable of recapitulating Aβ and tau pathologies when expressing fAD causing mutations [19,20] paves the way for the development of an interesting platform to study AD-like pathology induced by HSV-1 in an organoid context. The establishment of this novel experimental system will enable us to study the pathophysiological mechanisms linking AD with HSV-1 infection. Finally, broadening the knowledge on both phenomena could contribute to a better understanding of AD and the identification of new biomarkers and therapeutic targets, thereby promoting advances in translational and clinical fields of such a devastating disease.

## Figures and Tables

**Figure 1 microorganisms-11-01205-f001:**
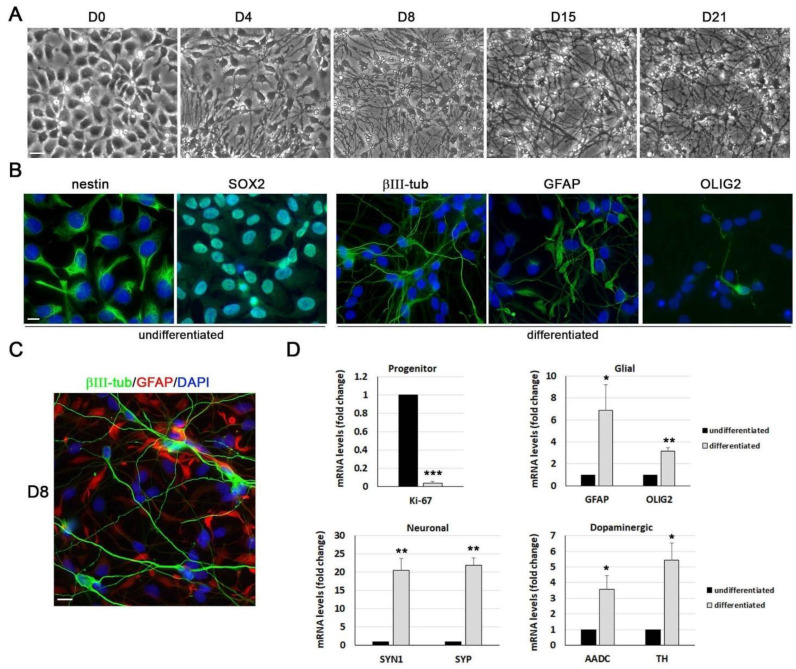
Differentiation of ReNcell VM cells into neurons and glial cells. (**A**) Phase-contrast microscopic images of the differentiation process in ReNcell VM cells. Undifferentiated cells (D0) and cells at day 4, 8, 15, and 21 of differentiation are shown. Scale bar: 50 µm. (**B**) Immunofluorescence images of undifferentiated and 8-day differentiated cells using antibodies specific for progenitor (Nestin and SOX2), neuron (βIII-tub), astrocyte (GFAP), and oligodendrocyte (OLIG2) markers (green). DAPI-stained nuclei are also shown (blue). Scale bar: 10 µm. (**C**) Immunofluorescence image of 8-day differentiated cells using antibodies specific for βIII-tub and GFAP. The merged image combines the βIII-tub (green), GFAP (red), and DAPI (blue) signals. Scale bar: 20 µm. (**D**) Analysis of gene expression of progenitor (Ki-67), neuronal (SYN1 and SYP), dopaminergic (AADC and TH), and glial (GFAP and OLIG2) markers by RT-qPCR in undifferentiated and 8-day differentiated cells. Graph data show the mean ± SEM of 4 independent experiments (one sample *t*-test; * *p* < 0.05; ** *p* < 0.01; *** *p* < 0.001).

**Figure 2 microorganisms-11-01205-f002:**
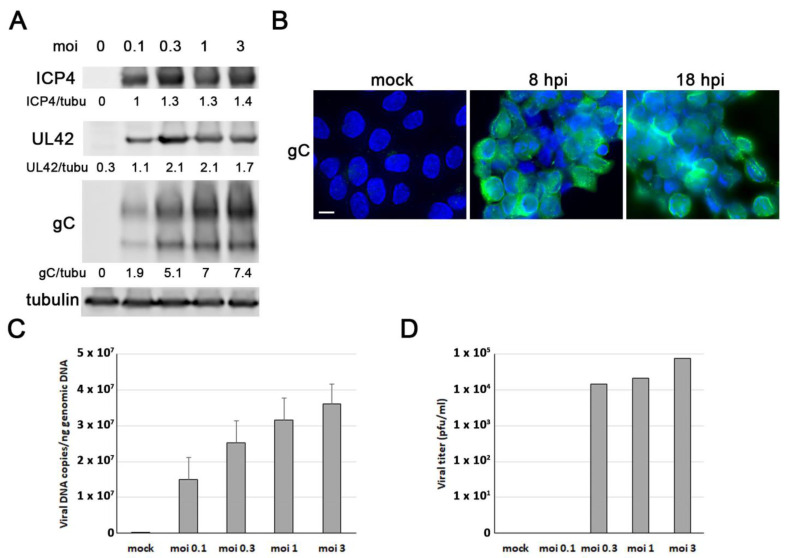
Characterisation of HSV-1 infection in ReNcell VM cells. (**A**) Western blot analysis of ICP4, UL42, and gC viral proteins in ReNcell VM cell lysates after an 18 h infection at different multiplicities of infection (moi). An α-tubulin blot to ensure equal loading is also shown. The ratio of viral proteins to α-tubulin, obtained by densitometric analysis, is shown below the blots. (**B**) Immunofluorescence images of ReNcell VM cells infected with HSV-1 at moi 3 for 8 and 18 h post-infection (hpi). Infection was monitored with an antibody specific to viral glycoprotein gC (green). Cellular nuclei were stained with DAPI (blue). Scale bar: 10 µm. (**C**) ReNcell VM cells were infected with different viral doses for 18 h, and the amount of viral DNA was analysed by quantitative PCR. The data represent the mean ± SD of two experiments performed in triplicate. (**D**) Extracellular viral titres were determined with plaque assays in ReNcell VM cells infected with HSV-1 at different mois for 18 h. Data of a representative experiment are shown.

**Figure 3 microorganisms-11-01205-f003:**
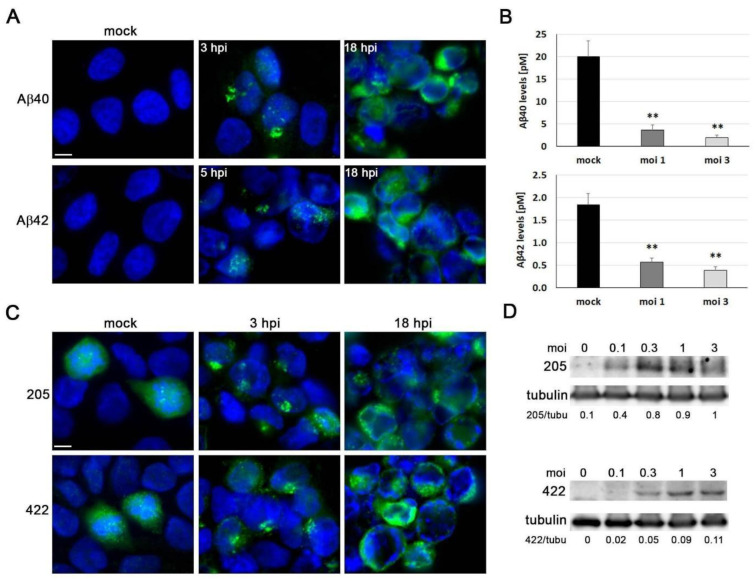
AD-like neurodegeneration markers induced by HSV-1 in ReNcell VM cells. (**A**) Study of intracellular accumulation of Aβ40 and Aβ42 peptides (green) by immunofluorescence in ReNcell VM cells noninfected (mock) or infected with HSV-1 at moi 3 for 3, 5, and 18 h. DAPI-stained nuclei are also shown (blue). Scale bar: 10 µm. (**B**) Quantification of secreted Aβ40 and Aβ42 levels by ELISA assays in ReNcell VM cells infected with HSV-1 at an moi of 1 and 3 for 18 h. The graph data represent the mean ± SEM of 4 independent experiments (Student’s *t*-test; ** *p* < 0.01). (**C**) ReNcell VM cells were exposed to HSV-1 at moi 3 for 3 and 18 h, and phosphorylated tau levels were assessed using the phosphorylation-sensitive antibodies thr205 and ser422 (green) with immunofluorescence assays. DAPI-stained nuclei are also shown (blue). Scale bar: 10 µm. (**D**) Western blot analysis of phosphorylated tau in ReNcell VM cell lysates after an 18 h infection at different mois. An α-tubulin blot to ensure equal loading is also shown. The ratio of phosphorylated tau to α-tubulin, obtained by densitometric analysis, is shown below the blots.

**Figure 4 microorganisms-11-01205-f004:**
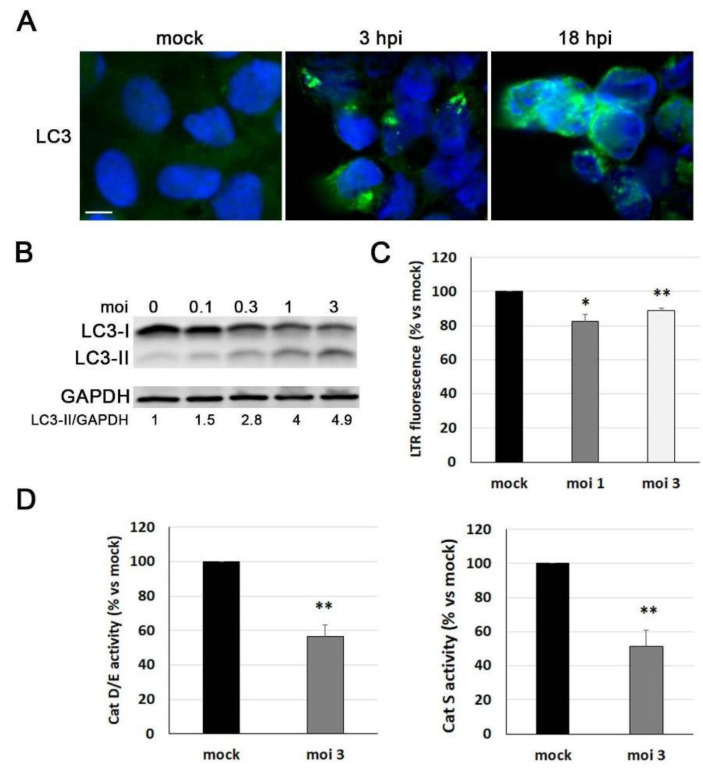
Lysosomal alterations induced by HSV-1 in ReNcell VM cells. (**A**) Immunofluorescence images of intracellular LC3 accumulation (green) in ReNcell VM cells infected with HSV-1 at moi 3 for 3 and 18 h. DAPI-stained nuclei are also shown (blue). Scale bar: 10 µm. (**B**) Western blot analysis of LC3-II in ReNcell VM cell lysates after an 18 h infection at different viral doses. A GAPDH blot to ensure equal loading is also shown. The ratio of LC3-II to GAPDH, obtained by densitometric analysis, is shown below the blots. (**C**) Analysis of lysosomal load by quantification of the LysoTracker Red (LTR) fluorescence in ReNcell VM cells infected with HSV-1 at an moi of 1 and 3 for 18 h. The graph data show the mean ± SEM of 4 independent experiments (one sample *t*-test; * *p* < 0.05; ** *p* < 0.01). (**D**) The relative enzymatic activities of cathepsins D/E and S were quantified in ReNcell VM cells infected at moi 3 for 18 h. Graph data show the mean ± SEM of 5 independent experiments (one sample *t*-test; ** *p* < 0.01).

**Figure 5 microorganisms-11-01205-f005:**
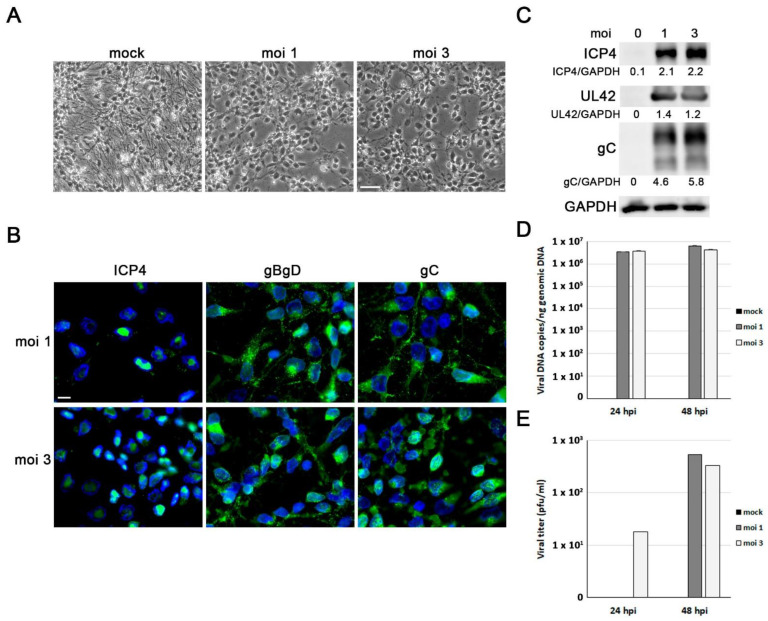
Characterisation of HSV-1 infection in differentiated ReNcell VM cells. (**A**) Phase-contrast microscopic images of ReNcell VM cells at day 8 of differentiation infected with HSV-1 at an moi of 1 and 3 for 24 h. Scale bar: 50 µm. (**B**) Immunofluorescence analysis of viral proteins ICP4, gBgD, and gC in 8-day differentiated ReNcell VM cells infected with HSV-1 at an moi of 1 and 3 for 18 h. Scale bar: 10 µm. (**C**) Analysis of ICP4, UL42, and gC levels by Western blot in 8-day differentiated ReNcell VM cells infected with HSV-1 at an moi of 1 and 3 for 18 h. The ratio of viral proteins to GAPDH is shown below the blots. Quantification of viral DNA levels using qPCR (**D**) and extracellular viral titres with plaque assays (**E**) were monitored in 8-days differentiated ReNcell VM cells infected with HSV-1 at an moi of 1 and 3 for 24 and 48 h. Data of a representative experiment are shown.

**Figure 6 microorganisms-11-01205-f006:**
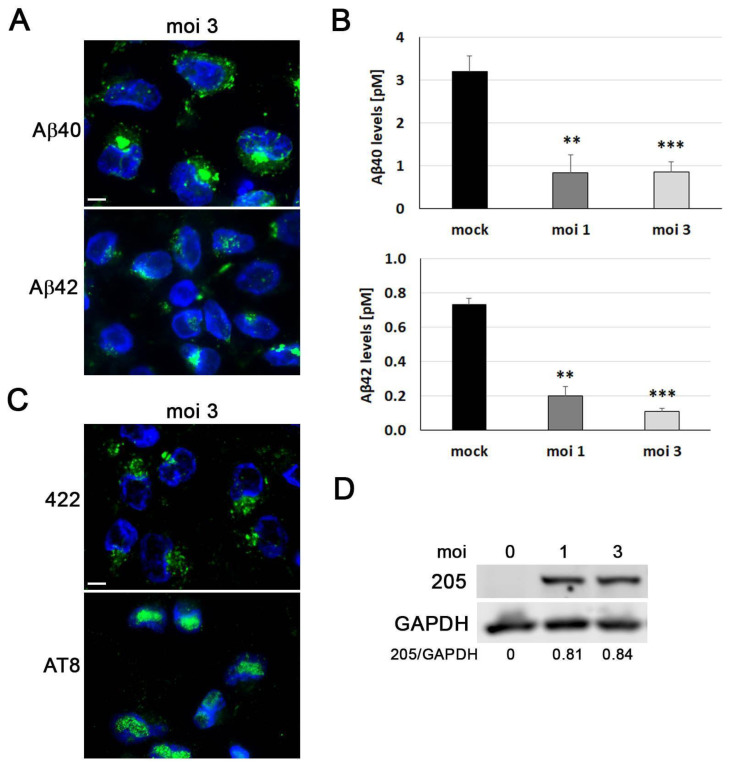
HSV-1 induces AD-like neurodegeneration markers in differentiated ReNcell VM cells. (**A**) After 8 days of differentiation, ReNcell VM cells were infected with HSV-1 at moi 3 for 18 h. The levels of Aβ40 and Aβ42 (green) were then measured using immunofluorescence experiments. DAPI-stained nuclei are also shown (blue). Scale bar: 10 µm. (**B**) Quantitative analysis using an ELISA of extracellular Aβ40 and Aβ42 levels in conditioned medium from 8-day differentiated ReNcell VM cells infected with HSV-1 at an moi of 1 and 3 for 18 h. The graph data represent the mean ± SEM of at least 3 independent experiments (Student’s *t*-test; ** *p* < 0.01; *** *p* < 0.001). (**C**,**D**) ReNcell VM cells at day 8 of differentiation were exposed to HSV-1 at an moi of 1 and 3 for 18 h, and the tau phosphorylated levels (green) were assessed using the phosphorylation-sensitive antibodies ser422 and AT8 using immunofluorescence (**C**) and thr205 with Western blot assays (**D**). DAPI-stained nuclei are also shown (blue). Scale bar: 10 µm. A GAPDH blot to ensure equal loading is also shown. The ratio of phosphorylated tau to GAPDH, obtained by densitometric analysis, is shown below the blots.

**Figure 7 microorganisms-11-01205-f007:**
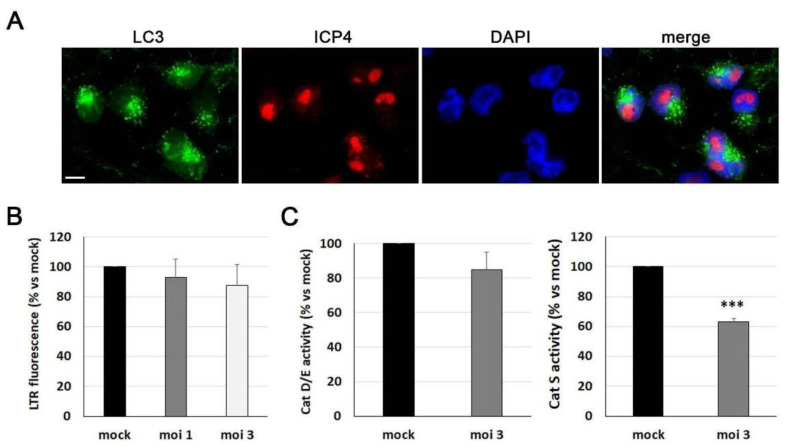
Lysosomal alterations induced by HSV-1 in differentiated ReNcell VM cells. (**A**) Immunofluorescence images of intracellular LC3 accumulation in 8-day differentiated ReNcell VM cells infected with HSV-1 at moi 3 for 18 h. Infection was monitored with an antibody specific to ICP4. DAPI-stained nuclei are also shown. Scale bar: 10 µm. (**B**) Lysosomal load was determined with the fluorescent lysosomotropic probe LTR in 8-day differentiated ReNcell VM cells infected with HSV-1 at an moi of 1 and 3 for 18 h. The graph data show the mean ± SEM of 4 independent experiments. (**C**) The results of lysosomal cathepsin activity assays were measured in ReNcell VM cells differentiated for 8 days and infected with HSV-1 at moi 3 for 18 h. The graph data show the mean ± SEM of 5 independent experiments (one sample *t*-test; *** *p* < 0.001).

**Table 2 microorganisms-11-01205-t002:** List of primers used in RT-qPCR analysis of ReNcell VM differentiation.

Gene *		Forward Primer	Reverse Primer
Proliferation	*MKI67*	5′-ATCGTCCCAGGTGGAAGAGTT-3′	5′-ATAGTAACCAGGCGTCTCGTGG-3′
Neuronal	*SYN1*	5′-TCAGACCTTCTACCCCAATCA-3′	5′-GTCCTGGAAGTCATGCTGGT-3′
	*SYP*	5′-CGAGGTCGAGTTCGAGTA CC-3′	5′-AATTCGGCTGACGAGGAGTA-3′
Dopaminergic	*TH*	5′-GCGCAGGAAGCTGATTGCTG-3′	5′-TGTCTTCCCGGTAGCCGCTG-3′
	*AADC*	5′-GAGTCACTGGTGCGCCAGGA-3′	5′-CCGTGCGAGAACAGATGGCA-3′
Astrocytes	*GFAP*	5′-CAACCTGCAGATTCGAGAAA-3′	5′-GTCCTGCCTCACATCACATC- 3′
Oligodendrocytes	*OLIG2*	5′-GCTGCGACGACTATCTTCCC-3′	5′-GCCTCCTAGCTTGTCCCCA-3′
Housekeeping	*ACTB*	5′-AGTGTGACGTGGACATCCGCAAAG-3′	5′-GTCCACCTTCCAGCAGATGTGGAT-3′

* *MKI67* (Ki-67); *SYN1* (synapsin I); *SYP* (synaptophysin); *TH* (tyrosine hydroxylase); *AADC* (dopamine decarboxylase); *GFAP* (glial fibrillary acidic protein); *OLIG2* (oligodendrocyte transcription factor); *ACTB* (β-actin).

## Data Availability

Not applicable.

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
