# Peer review of "Herpes Simplex Virus Type 1 Induces AD-like Neurodegeneration Markers in Human Progenitor and Differentiated ReNcell VM Cells"

_microorganisms, 2023, doi:10.3390/microorganisms11051205_

Round 1
Reviewer 1 Report
In the present manuscript, Salgado et al., investigated the use of the ReNcell VM cells, a human neural stem cell line, as a model to study “in vitro” the Herpes simplex virus type 1 (HSV-1) infection and of the possible effects on the development of molecular characteristics of Alzheimer's disease.
The authors confirm the infectivity of HSV-1 virus both in neural precursors and in differentiated cells. They also found that infection by this virus causes Alzheimer's-type neurodegeneration in infected cells (both progenitors and differentiated cells).
The main phenotypes caused were intracellular accumulation of AB peptides (AB40 and AB42), Tau phosphorylation, and alterations in lysosomal pathways.
The study itself is of great interest to the research field, and only a few minor points could be better addressed.
1-In the methods section, NeuN was included as a used marker, however, it is not present in the table and result section.
2-The ReNcell VM line is derived from the ventral mesencephalon, it is supposed to generate dopaminergic neurons, What is the percentage of TH+ neurons generated? And what is the percentage of the different phenotypes generated (neurons, glia, oligodendrocytes)?.
3-In discussion section, …ReNcell VM line is not derived from human pluripotent stem cells. This is an immortalized human neural stem cell line derived from ventral mesencephalon (Donato et al., 2007).
Reviewer 2 Report
The present study describes Herpes simplex virus type 1 induces AD-like neurodegeneration markers in human progenitor and differentiated ReNcell 3 VM cells. The study is well conducted and provides new information regarding the interaction of the Herpes simplex virus and ReNcell 3 VM cells. The study can be considered for publication after the authors address the following observations:
- Please demonstrate the mechanism by which HSV-1 infection decreases the lysosomal load in RenCell 3 VM cells.
- Please demonstrate the mechanism by which HSV-1 infection results in AB40 and AB42 accumulation in the cell. Is this behavior related to Alzheimer´s disease?
- Please review the entire manuscript and correct minor misspellings.
